# Potential Antagonistic Effects of Acrylamide Mitigation during Coffee Roasting on Furfuryl Alcohol, Furan and 5-Hydroxymethylfurfural

**DOI:** 10.3390/toxics7010001

**Published:** 2018-12-21

**Authors:** Dirk W. Lachenmeier, Steffen Schwarz, Jan Teipel, Maren Hegmanns, Thomas Kuballa, Stephan G. Walch, Carmen M. Breitling-Utzmann

**Affiliations:** 1Chemisches und Veterinäruntersuchungsamt (CVUA) Karlsruhe, Weissenburger Strasse 3, 76187 Karlsruhe, Germany; jan.teipel@cvuaka.bwl.de (J.T.); maren.hegmanns@cvuaka.bwl.de (M.H.); thomas.kuballa@cvuaka.bwl.de (T.K.); stephan.walch@cvuaka.bwl.de (S.G.W.); 2Coffee Consulate, Hans-Thoma-Strasse 20, 68163 Mannheim, Germany; schwarz@coffee-consulate.com; 3Chemisches und Veterinäruntersuchungsamt Stuttgart, Schaflandstr. 3/2, 70736 Fellbach, Germany; Carmen.Breitling-Utzmann@cvuas.bwl.de

**Keywords:** coffee, acrylamide, furfuryl alcohol, furan, 5-hydroxymethylfurfural, risk assessment

## Abstract

The four heat-induced coffee contaminants—acrylamide, furfuryl alcohol (FA), furan and 5-hydroxymethylfurfural (HMF)—were analyzed in a collective of commercial samples as well as in *Coffea arabica* seeds roasted under controlled conditions from very light Scandinavian style to very dark Neapolitan style profiles. Regarding acrylamide, average contents in commercial samples were lower than in a previous study in 2002 (195 compared to 303 µg/kg). The roasting experiment confirmed the inverse relationship between roasting degree and acrylamide content, i.e., the lighter the coffee, the higher the acrylamide content. However, FA, furan and HMF were inversely related to acrylamide and found in higher contents in darker roasts. Therefore, mitigation measures must consider all contaminants and not be focused isolatedly on acrylamide, specifically since FA and HMF are contained in much higher contents with lower margins of exposure compared to acrylamide.

## 1. Introduction

Acrylamide is a heat-induced contaminant with frequent occurrence in foods and beverages [1,2,3,4]. It has been classified by the International Agency for Research on Cancer (IARC) as probably carcinogenic to humans (group 2A) [5]. The EFSA suggested that its margin of exposure indicates a concern for neoplastic effects based on animal evidence [6]. Coffee is an important topic in reduction of acrylamide, because its consumption may lead to 20–30% of total daily intake [7].

Following the first findings of acrylamide in foods and research into its formation mechanism [8,9], it was quickly discovered that coffee behaves differently from all other foods. While typically, the acrylamide content rises with color or browning degree due to its origin as a Maillard reaction product, for coffee, its content decreases from light to very dark roasts [10]. The maximum of acrylamide is formed very early in the roast and then decreases until the desired roasting degree is reached. Experimental studies have shown that the final acrylamide content purely depends on the roasting degree but not on the profile by which this degree is achieved (i.e., neither very slow nor very quick roasting methods have any influence) [10]. Currently, literature offers only speculation into the breakdown product of acrylamide during roasting or the reaction leading to its degradation [11].

Acrylamide is a product formed during coffee roasting by the Maillard reaction, a major pathway comprising the reaction between asparagine and reducing sugars [12,13]. The formation capacity is limited by the amount of asparagine [14], which is the reason for higher acrylamide contents found in *Coffea canephora* (“robusta”) coffee due to its higher asparagine content.

Mitigation options may start with agronomy (e.g., species and variety selection, fertilization etc.) and roasting, but have also included strategies during processing such as asparaginase addition or lactic acid bacteria, none of which left the feasibility stage [15]. Careful removal of defective coffee beans is recommended, because these contain significantly higher amounts of asparagine (>2 fold), which is a major precursor of acrylamide formation [7,16]. Storage of coffee may lead to considerable reduction, but the final brew preparation is believed to have little influence due to the excellent water-solubility of acrylamide [15]. Some authors suggested that the variation detected in commercial samples may predominantly reflect differences in storage time [17]. Supercritical fluid extraction can be applied to reduce acrylamide by up to 79% [18]. Vacuum processing was suggested as a measure to reduce acrylamide in medium roasted coffee by 50% [19].

From all these factors, roasting was the predominant focus of previous research, and consistent findings hint that an increased roasting degree leads to a decrease in acrylamide formation [10,14,20,21,22,23,24].

Following several years of voluntary industry action with minimization concept [25], mitigation measures and benchmark levels for the reduction of the presence of acrylamide in food were recently implemented in an EU regulation [26]. The producers need to identify the critical roast conditions to ensure minimal acrylamide formation. They also need to ensure that the level of acrylamide in coffee is lower than the benchmark level of 400 µg/kg.

Besides acrylamide, coffee may contain further heat-induced contaminants that were also classified by IARC. Namely, furan [27], and furfuryl alcohol (FA) [28,29] are possibly carcinogenic to humans (group 2B). For 5-hydroxymethylfurfural (HMF), some evidence of carcinogenic activity was found in animal experiments [30,31], but the compound has not yet been classified by IARC. Out of these, furan is the compound in coffee studied most intensely, including large surveys [32,33,34], while less research is available on furfuryl alcohol [35,36,37,38] and HMF [39].

## 2. Materials and Methods

### 2.1. Analytical Methodology

The analysis of acrylamide was conducted according to the standard method EN 16618:2015 using liquid chromatography in combination with tandem mass spectrometry (LC/MS/MS) [40]. In deviation to this standard, samples were defatted with a mixture of isohexane and butyl methyl ether. Furthermore solid-phase extraction (SPE) was only used for clean-up, not for concentrating the acrylamide [11]. With this method, a limit of detection (LOD) of 10 µg/kg, and a limit of quantification (LOQ) of 30 µg/kg can be achieved. A repeatability relative standard deviation (RSDr) of 6% was determined within our laboratory. The method was applied successfully in several proficiency tests.

Analysis of furan was conducted using headspace-GC-MS and quantification with internal standard (furan-d_4_) as previously described [41]. A multipoint calibration (0.65–12.94 mg/kg) was used for quantification in SIM-Mode on a GC 7890B with MSD 5977B (Agilent Technologies, Waldbronn, BW, Germany) instead of the previously used standard addition. With this method, a LOD of 0.36 mg/kg and a LOQ of 1.2 mg/kg was achieved (0.5 g coffee sample weight). A RSDr of 3.5% was determined within our laboratory.

Analysis of furfuryl alcohol (FA) and 5-hydroxymethylfurfural (HMF) was accomplished using nuclear magnetic resonance (NMR) spectroscopy as previously described [42]. The within-laboratory RSDr was 6% for FA and 8% for HMF. LOD and LOQ were 12 and 39 mg/kg for FA and 6 and 23 mg/kg for HMF, respectively.

### 2.2. Samples and Roasting Experiments

Samples were obtained from official sampling for food control purposes in the German federal state Baden-Württemberg from all stages of trade, mainly supermarkets and artisanal roasters. For roasting experiments, two directly imported single estate terrace coffees (*Coffea arabica* and *canephora*) were supplied by Amarella Trading (Mannheim, BW, Germany).

Twelve separate 2.4 kg batches of coffee beans were roasted using an FZ-94 Laboratory Roaster (CoffeeTech, Tel Aviv, Israel). Roasting was conducted using either pure *Coffea arabica* or pure *Coffea canephora* samples. The roasting profiles (e.g., regarding temperature endpoints) were based on expert roasters’ experience as best suitable for the intended coffee roast type. The systematically different roast profiles were recorded and controlled using Artisan v1.5.0 (Artisan-Scope.org, Poing, BY, Germany, 2018, https://artisan-scope.org).

### 2.3. Risk Assessment Methodology and Statistics

Risk assessment was conducted using the margin of exposure (MOE) methodology according to the method for comparative risk assessment previously published for alcoholic beverages [3]. Statistical correlations were assessed using linear regression analysis calculated with OriginPro V7.5 (OriginLab Corporation, Northampton, MA, USA) with *R* being the correlation coefficient and *p* being the significance of Pearson’s test for linear relation. *p* values below 0.05 are assumed as being significant.

## 3. Results

### 3.1. Results of Roasting Experiments

Two green coffee samples (*Coffea arabica and canephora*) were subjected to roasting using six different profiles, namely coffee roasting (quick and slow drying), espresso roasting (quick and slow drying) as well as Scandinavian roasting (very light roasting) and Neapolitan roasting (very black roasting). The roasting profiles for the *C. canephora* roasting are shown in Figure 1. Profiles for *C. arabica* roasting were similar (data not shown).

Some numerical descriptors of the roasting profiles are provided in Table 1 as well as the analytical results for the samples. The individual roasting profile had a significant influence on the contents of the process contaminants. The area under the curve (AUC) is inversely related to acrylamide content (*R* = −0.59; *p* = 0.045; *n* = 12), while the contents of furfuryl alcohol (*R* = 0.78; *p* = 0.003; *n* = 12) and furan (*R* = 0.63; *p* = 0.027; *n* = 12) are positively correlated to this roasting parameter, independent of the coffee species. Furan (*R* = 0.65; *p* = 0.021; *n* = 12) and furfuryl alcohol (*R* = 0.82; *p* = 0.001; *n* = 12) are significantly positively correlated to drop temperature. The other parameters were not significantly correlated with any analyte.

There is an inverse linear statistically significant relationship between acrylamide and furfuryl alcohol (*R* = −0.85; *p* < 0.001; *n* = 12), and between acrylamide and HMF (*R* = −0.73; *p* = 0.007; *n* = 12). None of the other pairs for contaminants were significantly correlated; however, in tendency, acrylamide and furan were also inversely correlated, while furfuryl alcohol is positively correlated with HMF and furan.

### 3.2. Results of Commercial Sample Analyzes

The full results of analysis are provided in Appendix A, Table A1. The results are summarized in Table 2. From the sub-group of samples analyzed for both acrylamide and furfuryl alcohol, an inverse linear relationship was detected (*R* = −0.59; *p* = 0.008; *n* = 19). However, no correlation between HMF and acrylamide was detected, while HMF and furfuryl alcohol were positively correlated (*R* = 0.50; *p* = 0.007; *n* = 28). The data set of furan analysis was too small for meaningful statistical analysis.

Despite the low number of samples, the comparison of results in Table 2 suggests that the acrylamide content in roasted coffee and in instant coffee may have decreased over the years. None of the samples has exceeded the new EU benchmark levels.

### 3.3. Comparative Risk Assesment of Heat-Induced Contaminants in Coffee

Finally, the results of comparative risk assessment using the margin of exposure methodology are shown in Table 3. The risk assessment uses survey data from the literature due to the restricted, non-representative sampling in the current study.

While the contents of acrylamide and furan are much lower than the contents of furfuryl alcohol and HMF, the toxicity thresholds of both compounds are also much lower, with acrylamide being the compound showing effects at the lowest concentration of all four compounds. Nevertheless, due to the higher exposure, HMF and furfuryl alcohol have the lowest margins of exposure. Three of the compounds, acrylamide, furfuryl alcohol and HMF, have MOEs below 10,000. Furan falls below this threshold only in worst-case scenarios (P95 exposure) and can be seen as a compound with lower risk. However, HMF is believed to operate by a non-genotoxic mechanism and hence an uncertainty factor of 100 (instead of 10,000 for genotoxic compounds) may be sufficient to exclude public health concerns.

## 4. Discussion

Roasting properties of coffee are basically dependent on the amount of heat transferred into the coffee beans during roasting and on the roasting time [17]. A good indicator for the achieved heat transfer rate is the area under the curve of the roasting profile. These values show a negative correlation with acrylamide during our roasting experiment, confirming the inverse relationship of roasting energy and acrylamide [10,14,20,21,22,23,24]. In contrast, the other contaminants under study (furfuryl alcohol, furan and HMF) appear to be positively related to the roasting energy, meaning the highest contents are typically found in the strongest roasts (espresso).

Interestingly, despite early findings that acrylamide in coffee decreases with the roasting degree, there is still considerable misinformation about this topic. Some small artisanal coffee roasters even advertise on their webpages that their “mild” roasting process with temperatures rising only up to 200 °C would result in lower acrylamide contents. The contrary being clearly the case, however.

Compared to results from our institutes published in 2002 (average acrylamide content in coffee: 303 µg/kg, median 313 µg/kg; 90% percentile 461 µg/kg) [11], the contents found during this study were lower. In Germany, the minimization of acrylamide has been most advanced of all EU member states [25]. Manufacturers should therefore not be challenged, even if the current benchmark level should become the new legal maximum limit [25]. Our results confirm this assumption, since none of our official samples exceeded the current benchmark level.

Some authors have questioned the influence of species, e.g., Mojksja and Gielecinska [22], who found no significant difference in acrylamide contents between Arabica and Robusta coffee. Our restricted results of two pure *C. canephora* coffees (260–270 µg/kg) lie actually above the average acrylamide contents of all coffee samples (196 µg/kg), which is consistent with the majority of literature [14,24,48]. However, in our case a comparison is confined due to the fact that the species is unknown in most of the analyzed commercial samples. It may be speculated that the difference is caused by the lower quality of commercial *C. canephora* coffee with a higher degree of defective beans. However, the comparison of our high-quality single estate terrace coffees (Table 1) also points to higher levels of acrylamide in *C. canephora*.

There are only few studies available on the correlation of other contaminants with acrylamide. Kocadagli et al. [49] studied the kinetics of both acrylamide formation and HMF formation and found similar tendencies, meaning both acrylamide and HMF are reduced by more intense roasts. This is in contrast to our results, which detected this behavior only for acrylamide but not for HMF. An explanation may the different methodology in Kocadagli et al. [49], which did not apply a commercial coffee roaster but only an oven at 220 °C for 5–60 min. We therefore believe that our results may have a higher relevance for commercial coffee roasting. Nevertheless, there remains some uncertainty in HMF exposure from coffee. For example, the survey reported by Arribas-Lorenzo [39] from Spain found higher HMF levels than our study with less samples. According to the German Federal Institute for Risk Assessment (BfR) evaluation, the levels of HMF in foods were suggested to exhibit no identifiable health risk for the consumer [50]. However, the BfR did not include coffee in its evaluation of HMF due to a lack of food monitoring data necessary for exposure assessment.

For other heat-induced contaminants besides acrylamide, no action has been typically taken to reduce levels and there are also no EU benchmark or maximum levels for heat-induced contaminants besides acrylamide. Therefore, focus and research activity have been mainly aimed at acrylamide in the past. The Codex Code of Practice to reduce acrylamide in foods currently does not provide guidance for coffee because to date “no commercial measures for reducing acrylamide in coffee are currently available” [15,51]. While this opinion is probably outdated, as various measures have shown to be effective (see introduction), our findings suggest that indeed no measures should be implemented that solely focus on acrylamide. Using a holistic risk assessment approach, all major heat-induced contaminants in coffee need to be modelled prior to pointing out any measure. Otherwise it could well mean that the benefit gained by reduction of acrylamide might be outweighed by the elevated risk of other contaminants such as furfuryl alcohol that are concomitantly increased by the applied measure. As other authors have shown [7,20], holistic risk-benefit analysis would be most preferable as the mitigation of acrylamide might not only lead to increased formation of other contaminants such as furfuryl alcohol [36], but may also lead to reduced contents in beneficial compounds in coffee such as antioxidants.

Compared to other lifestyle factors such as tobacco smoking or alcohol drinking, the cancer risk from coffee (if any exists) appears to be rather low. According to IARC, epidemiological studies even suggest a lack of carcinogenicity of drinking coffee for cancer of the liver [52,53], which is the major target organ of heat-induced contaminants. Bladder cancer was the only cancer site for which an increased risk had been observed in some earlier epidemiological studies, leading to an IARC grouping as 2B in 1991 [54]. However, more recent well-conducted epidemiologic studies were unable to replicate the association with bladder cancer, and coffee consumption has been removed from the classification as a possible/probable human carcinogen [52,53].

## Figures and Tables

**Figure 1 toxics-07-00001-f001:**
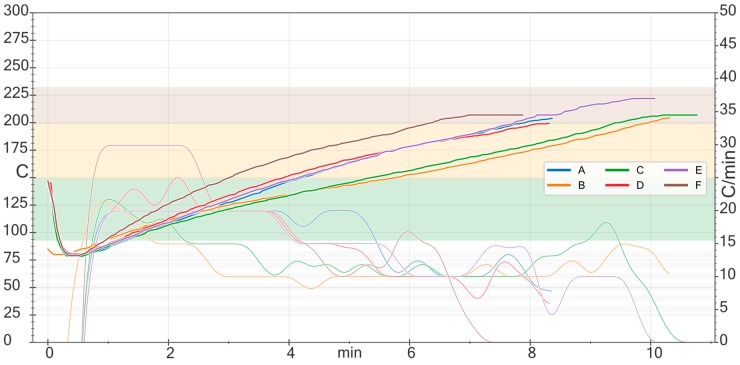
Profiles of experimental coffee roasting (A: Coffee quick drying; B: Coffee slow drying; C: Espresso slow drying; D: Scandinavian coffee; E: Espresso Neapolitan; F: Espresso quick drying).

**Table 1 toxics-07-00001-t001:** Indicators of roasting (data for *C. canephora* roast; *C. arabica* data similar) and analytical results of roasted coffee (*C. arabica*/*C. canephora*).

Profile	Charge ^1^ [°C]	Drop ^2^ [min]	Drop ^2^ [°C]	AUC ^3^ [°C·min]	Acrylamide [µg/kg]	Furfuryl Alcohol [mg/kg]	Furan [mg/kg]	HMF [mg/kg]
Scandinavian coffee	145	08:22	200	555	470/480	70/93	<1.2/2.5	40/47
Coffee quick drying	140	08:24	204	566	200/390	124/94	1.7/2.7	74/49
Coffee slow drying	85	10:21	205	673	210/420	128/92	1.5/2.6	62/43
Espresso quick drying	147	07:55	203	625	170/300	170/117	2.5/4.9	66/47
Espresso slow drying	140	10:48	207	762	150/290	173/133	2.6/5.0	78/42
Neapolitan espresso	145	10:06	222	796	130/250	223/189	3.6/7.6	84/32

^1^ Temperature at charge of roaster. ^2^ Drop = end of roast. ^3^ Area under the curve (indicator how much total energy the beans have received during roasting).

**Table 2 toxics-07-00001-t002:** Comparison of acrylamide analysis results from 2002 with current results (summary from Annex A, Appendix A).

Category according to EU Regulation 2017/2158	Year of Analysis	Number of Samples	Average [µg/kg]	Median [µg/kg]	90th Percentile [µg/kg]
Roast coffee	2002 (data from [11])	5	303	313	461
Roast coffee	2015	4	118	130	138
Roast coffee	2018	22	195	165	306
Instant (soluble coffee)	2013	6	642	686	831
Instant (soluble coffee)	2015	7	483	356	805
Instant (soluble coffee)	2016	5	379	269	664
Instant (soluble coffee)	2018	13	555	600	842
Coffee substitutes exclusively from cereals	2013–2018	6	401	418	563
Coffee substitutes from a mixture of cereals and chicory	2012–2018	16	587	525	805

**Table 3 toxics-07-00001-t003:** Risk assessment of several roasting contaminants in coffee.

Contaminant	Average/P95 Content in Roasted Coffee	Average/P95 Exposure for Drinking 1 Cup of Coffee ^1^	Toxicological Threshold ^2^	Average/P95 Margin of Exposure (MOE) ^3^
Acrylamide	249/543 µg/kg [6]	0.05/0.10 µg/kg bw/day	0.18 mg/kg bw/day (BDML10) [43]	3800/1700
Furfuryl alcohol	251/392 mg/kg [35]	0.05/0.07 mg/kg bw/day	53 mg/kg bw/day (NOEL) [44]	1110/710
HMF	689/1688 mg/kg [39]	0.13/0.32 mg/kg bw/day	79 mg/kg bw/day (BMDL10) [30]	600/250
Furan	38/107 µg/L [33]	0.12/0.14 µg/kg bw/day [33]	1.28 mg/kg bw/day (BMDL10) [45]	42,134/3113 [33]

^1^ Calculated assuming 14 g of coffee powder per 0.2 L cup (according to ISO 6668 [46]) and assuming 100% extraction yield, except for furan for which data from brewed beverage analyses were available. Average bodyweight 73.9 kg [47]. The data for furan were probabilistically calculated and taken from [33]. All other values were own calculations using point estimates. ^2^ NOEL: no-observed effect level; BMDL10: benchmark dose lower confidence limit for 10% response. ^3^ MOE = Toxicological threshold/exposure. Values pessimistically rounded to significance. The higher the MOE, the lower the risk. A MOE > 10,000 is typically interpreted as low risk for genotoxic carcinogens, while >100 is used for non-genotoxic compounds with thresholded effects.

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
