# Peer review of "Potential Antagonistic Effects of Acrylamide Mitigation during Coffee Roasting on Furfuryl Alcohol, Furan and 5-Hydroxymethylfurfural"

_toxics, 2018, doi:10.3390/toxics7010001_

Reviewer 1 Report

In 2011 the BfR published a statement that HMF is a compound of very low risk; this should be included in the references and it is questionable if HMF could be denominated as a contaminant

It is not stated how the endpoint of roasting was defined; this should be stated clearly.

Labels in Fig.1 are very small and difficult to read.

Tabel 1: units of AUC are missing.
When using NMR for quantification the measurement error is ca. 5 % this should also be reflected in the values given.

Table 2: From the capture it is not clear what was measured.

Table 3: From an exposure of 0.05 the MOE of 3815 was calculated; the authors should think of the concept of significance when giving values, it would be of great help to consider the error propagation law when estimating errors of measured values.

line 185ff: it should be noted that coffee consumption increases the risk of bladder cancer and in smokers the risk of lung cancer!

Recent publications on furfuryl alcohol (e.g. Albouchi and co-workers) should be included.

Author Response

In 2011 the BfR published a statement that HMF is a compound of very low risk; this should be included in the references and it is questionable if HMF could be denominated as a contaminant

The BfR publication was included in the reference list as requested. However, HMF is clearly belonging to the group of heat-induced contaminants according to Code Alimentarius or EU definitions, as it is a compound not intentionally added to foods, but formed inadvertently during food processing. 

It is not stated how the endpoint of roasting was defined; this should be stated clearly.

The endpoints were defined on empirical knowledge from expert roasters experience. This information was added to the materials section.

Labels in Fig.1 are very small and difficult to read.

We were unable to retroactively change the font sizes in the Artisan software, but believe that the labels are well readable in A4 printout or on screen.

Tabel 1: units of AUC are missing.

The unit was added to the table.

When using NMR for quantification the measurement error is ca. 5 % this should also be reflected in the values given.

Validation results for NMR were included as requested.

Table 2: From the capture it is not clear what was measured.

Thanks your for spotting this mistake. Acrylamide was added to the caption.

Table 3: From an exposure of 0.05 the MOE of 3815 was calculated; the authors should think of the concept of significance when giving values, it would be of great help to consider the error propagation law when estimating errors of measured values.

Yes. For this reason average and P95 scenarios are included, as it is typically done in such risk assessments. The data appeared not sufficient for further analyses (e.g. using Monte Carlo methods) as it has been done in Ref. 33 for Furan. Our calculated MOE values in table 3 were pessimistically rounded to significance as requested.

line 185ff: it should be noted that coffee consumption increases the risk of bladder cancer and in smokers the risk of lung cancer!

Actually, no! This is outdated knowledge. It is of note that IARC has removed the bladder, which was the only remaining cancer site for coffee, from its evaluation, and downgraded coffee in its cancer classification from 2B to “not classifiable”. More recent epidemiological studies were unable to replicate the influence of coffee on the bladder, which was likely caused by confounding in earlier studies. The interaction with smoking is also not consistently found in epidemiological studies and may also be due to confounding in some earlier studies. Actually, the earlier finding of bladder cancer due to coffee is currently believed to have been caused by insufficient control for smoking in the statistical evaluations. We have added a remark on IARC cancer evaluations to the text.

Recent publications on furfuryl alcohol (e.g. Albouchi and co-workers) should be included.

Done.

Reviewer 2 Report

This manuscript describes the acrylamide, furfuryl alcohol and 5-hydroxmethylfurfural content in coffee beans following different types of roasting and in different types of commercial products.

This reviewer has major concerns about this manuscript.  Most specifically is the lack of details and description in all parts of the manuscript.  The lack of this information has made this review extremely difficult. 

 Although FA and HMF are included in the title, there is no mention of these chemicals or their significance in the introduction. 

The authors claim that roasting temperature along with many other factors may influence acrylamide content but still compare results from the present study (2012-2018) to previous studies conducted as early as 2002.  They also state that the average content was lower in the present studies but it is unclear as to the exact conditions and extraction methods that were used for the earlier studies and how they might influence acrylamide content.  Additionally, the numbers were based on an n value of 5 for the earlier studies compared to 19 for the present study and there is no mention of SE or SD.

There is no mention of toxicological aspects of the chemicals under investigation in the manuscript.

The authors make claims that FA and HMF were inversely related to acrylamide and found in higher contents in darker roast but there are no statistics to support this statement.  No n value is provided for these studies.

  The methods are lacking in detail.  The n value needs to be provided for the roasting studies.  Number of samples, types of samples, sources should be included.  Additional information on the roasting procedure needs to be included.  There is no mention of any type of statistical analysis.

It is not clear what is being measured in Figure 1.  More detail needs to be included in the legend.  Was this just from one sample/bean source? 

Table 1 does not indicate n value or SE

The results section of the manuscript should be separated into roasting experiments and commercial sample sections.  The transition between the two experiments is confusing and the manuscript does not flow well.

It is not clear how accurate statements can be made about the concentrations of acrylamide between such diverse samples (instant coffee to cereals).  Was the extraction method and analysis of acrylamide the same for all samples being analyzed?  Why was content of FA and HMF only analyzed for select samples?

The title for Table 2 states date range from 2015-2018 but data in Table S1 is from 2012-2018

Author Response

This manuscript describes the acrylamide, furfuryl alcohol and 5-hydroxmethylfurfural content in coffee beans following different types of roasting and in different types of commercial products.

This reviewer has major concerns about this manuscript.  Most specifically is the lack of details and description in all parts of the manuscript.  The lack of this information has made this review extremely difficult. 

 Although FA and HMF are included in the title, there is no mention of these chemicals or their significance in the introduction. 

An introductory paragraph on FA, HMF and furan was added. However, the focus of the paper has been acrylamide and not the other compounds.

The authors claim that roasting temperature along with many other factors may influence acrylamide content but still compare results from the present study (2012-2018) to previous studies conducted as early as 2002.  They also state that the average content was lower in the present studies but it is unclear as to the exact conditions and extraction methods that were used for the earlier studies and how they might influence acrylamide content.  Additionally, the numbers were based on an n value of 5 for the earlier studies compared to 19 for the present study and there is no mention of SE or SD.

As these were commercial samples, as stated in the text, processing and roasting conditions are unknown. Thus it the reasons for the decrease of acrylamide in the samples can only be speculated upon, except for the fact that roasters were completely unaware about the problem and no mitigation effort had been started in 2002. Some information into the statistical distribution is provided by specifying average, median and P95. As no further data was available from the 2002 paper, we decided to use the same statistical data as specified in 2002. The text was changed to point out the mentioned problems in the data collective.

There is no mention of toxicological aspects of the chemicals under investigation in the manuscript.

References to toxicological reviews (e.g. IARC) were added for all compounds.

The authors make claims that FA and HMF were inversely related to acrylamide and found in higher contents in darker roast but there are no statistics to support this statement.  No n value is provided for these studies.

Actually, all raw data on which the statistical analysis was based is provided in the data appendix table, and p-values were provided. N-values were added as requested.

The methods are lacking in detail.  The n value needs to be provided for the roasting studies. 

There were 2 studies: one on arabica and one on canephora. This was clarified in methods.

Number of samples, types of samples, sources should be included.  Additional information on the roasting procedure needs to be included.  There is no mention of any type of statistical analysis.

 The information is provided in sections 2.2. As the full roasting profile is provided, we fail to see what more could be included.

It is not clear what is being measured in Figure 1.  More detail needs to be included in the legend.  Was this just from one sample/bean source? 

Each curve is for the roasting of one batch (2.4 kg) as specified in section 2.2.

Table 1 does not indicate n value or SE

N=1 so no SE available. It must be considered that one sample analysed using three different techniques (LC/MS/MS, GC/MS and NMR) costs nearly 1,000 EUR, so the whole table with 6 roastings of two species has cost about 12,000 EUR. As we had no funding for such research, further replicates were impossible due to economic reasons.

The results section of the manuscript should be separated into roasting experiments and commercial sample sections.  The transition between the two experiments is confusing and the manuscript does not flow well.

The reviewer is correct. The sub-titles to sections 3.2 and 3.3 were somehow lost. The text is now separated into section 3.1 (roasting) and 3.2 (commercial samples).

It is not clear how accurate statements can be made about the concentrations of acrylamide between such diverse samples (instant coffee to cereals).  Was the extraction method and analysis of acrylamide the same for all samples being analyzed?  Why was content of FA and HMF only analyzed for select samples?

The applied standard method EN 16618:2015 has been validated for all these matrices and comparable results are obtained in our experience. We have only developed the methods for FA and HMF analysis in 2018, so that only a sub-group of samples could be analysed for all compounds.

The title for Table 2 states date range from 2015-2018 but data in Table S1 is from 2012-2018

The 2002 data were for roasted coffee only, not instant coffee. So the statement was more or less correct, but we decided to delete this level of specificity from the table title to avoid confusion. The years should be self-evident from column 2 of the table.

Round  2

Reviewer 2 Report

The authors have addressed my comments